# New Fire-Retardant Open-Cell Composite Polyurethane Foams Based on Triphenyl Phosphate and Natural Nanoscale Additives

**DOI:** 10.3390/polym16121741

**Published:** 2024-06-19

**Authors:** Kirill Cherednichenko, Egor Smirnov, Maria Rubtsova, Dmitrii Repin, Anton Semenov

**Affiliations:** Department of Physical and Colloid Chemistry, National University of Oil and Gas «Gubkin University», 65 Leninsky Prospekt, Moscow 119991, Russia; cherednichenko.k@gubkin.ru (K.C.); egorsmirnovrgu@yandex.ru (E.S.); rubtsova.m@gubkin.ru (M.R.); d.repin2124@yandex.ru (D.R.)

**Keywords:** polyurethane foam, fire retardance, halloysite, nanocellulose, functional composites

## Abstract

Despite the mechanical and physical properties of polyurethane foams (PUF), their application is still hindered by high inflammability. The elaboration of effective, low-cost, and environmentally friendly fire retardants remains a pressing issue that must be addressed. This work aims to show the feasibility of the successful application of natural nanomaterials, such as halloysite nanotubes and nanocellulose, as promising additives to the commercial halogen-free, fire-retardant triphenyl phosphate (TPP) to enhance the flame retardance of open-cell polyurethane foams. The nanocomposite foams were synthesized by in situ polymerization. Investigation of the mechanical properties of the nanocomposite PUF revealed that the nanoscale additives led to a notable decrease in the foam’s compressibility. The obtained results of the flammability tests clearly indicate that there is a prominent synergetic effect between the fire-retardant and the natural nanoscale additives. The nanocomposite foams containing a mixture of TPP (10 and 20 parts per hundred polyol by weight) and either 10 wt.% of nanocellulose or 20 wt.% of halloysite demonstrated the lowest burning rate without dripping and were rated as HB materials according to UL 94 classification.

## 1. Introduction

Polyurethane foams (PUFs) are an indispensable class of lightweight materials that are widely used in modern industry and everyday life, from dishwashing sponges to thermal and acoustic insulation in construction, and from shoe materials and furniture fillers to molded parts in mechanical engineering, etc. [1,2,3,4]. Despite their excellent thermal and acoustic insultation properties [3,5], high chemical and mechanical stability [6,7], and low density and biological inertness [8], PUFs still have a major shortcoming, namely their high flammability [9]. The reasons for the high flammability of PUF can be attributed to three main factors. Firstly, the decomposition temperatures of the chemical bonds constituting the polyurethane polymer chain are relatively low (100–325 °C [10]). Secondly, the thermal decomposition and combustion of polyurethane result in the formation of flammable gases and free radicals, which in turn contribute to and accelerate combustion [9]. Finally, the porous structure of PUF provides the easy transfer of heat and a sufficient amount of oxygen, which are essential for maintaining the combustion process [11].

Over the past few decades, numerous studies have been performed to enhance PUF’s fire resistance [9,12,13,14,15,16]. Based on the findings of these studies, three main approaches can be distinguished: (1) the chemical modification of the polyurethane chain (i.e., the introduction of fire-resistant fragments/functional groups); (2) coating the PUF’s surface with a fire-resistant material (e.g., via the LbL technique or foam soaking); and (3) the incorporation of various flame retardants and other materials (including nanomaterials) during PUF synthesis. The latter approach implies the addition of a flame-retardant additive to one of the PUF’s components (usually polyol) prior to mixing with the second component and subsequent foam formation. Unlike the other two approaches mentioned above, this way of modifying PUF does not require the development of a new chemical formula for polyurethane and allows homogeneous foams of any form and volume to be obtained [9].

Halogen-containing compounds were a popular and widely applied fire retardant for PUF because of their comparatively low price and high effectiveness. Nevertheless, during combustion, they produce toxic gases that present a danger to human health and the environment [16,17,18]. To address this issue, a number of environmentally friendly halogen-free fire retardants based on nitrogen- and phosphorus-containing compounds have been elaborated [9,12,15,16,17,19,20]. The anti-flame activity of such fire retardants includes the emission of non-flammable gases and free radicals (e.g., PO_2_ and PO, etc.), which capture free radicals formed during urethane burning, thereby reducing or even completely terminating combustion [9,16]. Moreover, phosphorus-based flame retardants can enhance the carbonization of polyurethane, which in turn leads to the formation of a more stable char layer, inhibiting the heat transfer and pyrolysis of the remaining material [21].

It has been well established in the literature that the combination of various nanomaterials (e.g., carbon nanomaterials) with halogen-free fire retardants can enhance the overall fire resistance of PUF [9,11,12,15,16]. For example, the combination of functionalized graphene oxide (GO) with ammonium polyphosphate (APP) [22] or dimethyl methylphosphonate (DMMP) [23,24] increased char yield and, thus, provided a superior barrier effect compared to PUF modified with only a fire retardant. Synergetic effects between intumescents such as expandible graphite (EG) and various halogen-free fire retardants are also well documented phenomena [9]. For instance, the combination of EG with APP [25,26] resulted in a significant increase of limiting oxygen index (LOI) values. At elevated temperatures, EG forms a protective char layer that impedes the diffusion of heat and the flammable products of polyurethane decomposition within the foam. Additionally, EG expansion is usually accompanied by the emission of nonflammable gases (e.g., SO_2_, NO_2_, and CO_2_), diluting flammable gases [27]. Nevertheless, the production of fire-retardant composite PUF modified with synthetic carbon nanomaterials is limited by the relatively high cost of the latter.

In recent decades, growing ecological concerns and economic reasons have led researchers to employ cheap, environmentally friendly natural materials as nanoscale additives to fire resistant PUF [9,12,14]. Thus, several studies have demonstrated that costly carbon nanotubes can be effectively substituted for halloysite nanotubes (HNT) [28,29,30]. For example, to obtain flame-retardant PUF, Smith J.R. et al. proposed bilayer coatings based on HNT stabilized by branched polyethylenimine and polyacrylic acid. The deposition of three and five bilayers led to the significant enhancement of the foams’ flame-retardant properties. The findings of this study indicate that the combination of HNT and phosphorous-containing compounds produces a synergistic effect [28]. Wu F. et al. employed polysiloxane-modified HNT to modify PUF’s surface by dip-coating technique and obtained fire-resistant foams [29]. A recent study by the same research group examined the soaking of flexible foams by aqueous HNT suspension. The results indicated that the coating of PUF’s surfaces by even untreated HNT led to a notable improvement in fire resistance [30]. It is generally accepted in all the cases mentioned above that the outstanding mechanical properties of halloysite and its low thermal conductivity enabled HNT to participate in the formation of a more stable char layer, which in turn provided better flame and heat isolation. In addition to HNT, significant improvements in the fire resistance of PUF have also been demonstrated with the use of other nanoclays [31,32,33,34]. However, it should be underlined that these natural nanomaterials are currently commonly used as part of the flame-retardant coatings for PUF, which hinders the scale-up of flame-retardant PUF production.

Biopolymers are another important class of natural materials that can be good candidates for the modification of PUF. Cellulose is a biopolymer of considerable potential, exhibiting excellent mechanical properties, zero environmental impact, and a low production cost, largely due to its abundance in natural sources. In recent decades, this natural material has been successfully employed to enhance the mechanical characteristics, thermal properties, and sustainability of PUF [35,36,37,38,39,40,41]. In contrast to nanoclays, cellulose is typically incorporated into the foam structure via a blending with the polyol component prior to PUF synthesis (i.e., in situ synthesis). Nevertheless, a careful search of the literature on the application of cellulose or other biopolymers as PUF flame retardants yielded no results, with the exception of a report on the use of chitosan as a nitrogen-containing compound [9]. Consequently, the exploration of the flame-retardant properties of PUF modified by individual biopolymers (e.g., cellulose) and its combination with commercially available halogen-free fire retardants is of the utmost importance.

In the present work, natural aluminosilicate nanotubes and nanocellulose were applied individually and in combination with the commercially available phosphorous-containing fire retardant triphenyl phosphate (TPP) to enhance the fire retardance of open-cell PUF. These nanocomposite PUFs were obtained by an in situ polymerization method for the first time. The structure, compressibility, thermal stability, and flammability of the obtained foams were comprehensively investigated. A synergistic effect between TPP and two natural additives was observed during flame-retardancy studies.

## 2. Materials and Methods

### 2.1. Raw Materials

In this work, we used commercial open-cell PUF raw materials FlexFoam-iT!™ III (Smooth-On, Inc., Macungie, PA, USA). PUF raw materials were supplied in two components: component A (containing 4,4′-MDI CAS 101-68-8, MDI homopolymer CAS 25686-28-6, methylenebis(phenyl isocyanate) CAS 26447-40-5, benzene) and component B (containing polyol, catalysts, surfactants, and water as blowing agent). Triphenyl phosphate (TPP) was supplied by BLDpharm (98%; Shanghai, China), halloysite nanotubes (HNT) were supplied by Sigma Aldrich (St. Louis, MO, USA), and nanocellulose (NC) was obtained according to the procedure described previously [42].

### 2.2. Synthesis of Fire-Retardant Composite PUF

The reference PUF was synthesized according to the Smooth-On, Inc. manufacturer’s technical bulletin. An amount of 1.5 mL of component A was added to 3 mL of component B. The obtained mixture was manually mixed for 35 s (pot life) and then poured into the mold cast. The foam curing with backing pressure lasted 10 h, then the foam was retrieved and dried overnight.

To obtain the modified PUF, the fine dried powders of HNT, NC, and TPP were added to component B and thoroughly mixed. A typical TPP content in PUF usually varies from 10 to 20 php (parts per hundred polyol by weight) [43]. In this work, two different TPP contents were tested: 10 and 20 php. However, it should be noted that the mass fraction of HNT or NC was calculated with respect to the total mass of components A and B. To improve dispersion in the rather viscous component B, the obtained mixtures were treated in an ultrasonic bath for 5 min (the mixture temperature was kept around room temperature). Component A was then added to this mixture. The rest of the procedure was identical to the synthesis of the reference PUF. The formulations of all the synthesized PUFs are presented in Table 1. The PUF of each composition was obtained in batches of five samples. For convenience, the foams containing different amounts of TPP, HNT, and NC are referred to as PUF/TPP, PUF/HNT, and PUF/NC, respectively. The nanocomposite PUF samples containing TPP together with halloysite are referred to as PUF/TPP/HNT, while those containing TPP and cellulose are referred to as PUF/TPP/NC.

### 2.3. Characterization of Fire-Retardant Composite PUF

To investigate the structure of the obtained PUFs, the central part of their cross section was studied by light microscopy using an Olympus SZ61 stereomicroscope (Tokyo, Japan). The measurement of the PUF cells’ area was performed with the help of ImageJ 1.53t [44]. The obtained data were used to build the corresponding box plots illustrating the distribution of cell area using Origin Pro 8.1 software.

The investigation of HNT and TPP distribution in the PUF was carried out with help of scanning electron microscopy (SEM) and energy-dispersive X-ray spectroscopy (EDS). The PUF samples were fixed on SEM slabs and coated with thin carbon layer (20–30 nm) to avoid undesirable overcharging effects, and the corresponding artefacts in SEM micrograph. The acquisition of SEM micrographs was performed with the help of JEOL JIB 4501 multibeam system (Tokyo, Japan) at accelerating voltage of 15 kV. EDS measurements and acquisition of the corresponding elemental mappings was carried out with the help of a X-Max 20 mm^2^ Oxford Instruments EDS detector (Abingdon, UK) and INCA software V4.15.

To obtain the apparent density (ρ), the volume (*V*), and mass (*m*) of the samples presented by the series, five specimens of the same composition were measured. The apparent density was calculated from the *m*/*V* ratio.

The FTIR spectra of the synthesized PUF were acquired in the absorption mode in the 800–4000 cm^−1^ range using a Nicolet iS 10 FTIR Spectrometer equipped with a germanium ATR crystal. The spectral resolution of the device was 8 cm^−1^, and the acquisition time for each spectrum was 14 s. The obtained spectra were treated with the help of OMNIC Thermo Scientific and Origin Pro 8.1 software.

The compression force deflection of the synthesized nanocomposite foams were tested according to ASTM D3574 (Test C) [45]. In other words, the force required to achieve a 50% compression of the entire top area of the foam specimen was measured for three specimens of the same composition. According to protocol, each specimen was positioned on the lower support platen within the compression tester and subjected to a preliminary compression by means of a double lowering and raising of the upper platen to 75–80% of its original thickness at a rate of 250 mm/min. After a further 6 min without loading, the specimen was compressed by 50% of its thickness at a rate of 50 mm/min. The final force and displacement were determined after 60 s.

The thermal stability of the synthesized foams was probed with the help of thermogravimetric analysis (TGA) using a NETZSCH STA 449 F5 Jupiter synchronous thermal analyzer (Selb, Germany). Each measurement was carried out in an air atmosphere in a 30–800 °C temperature range at a constant heating rate of 10 °C/min. The thermogravimetric data obtained were treated with the help of NETZSCH Proteus 8.0 software and Origin Pro 8.1 software.

The flammability of the obtained PUF was tested using the horizontal burning test (UL 94-HB). In this test, a 2 cm flame was applied to the end of a horizontally placed PUF sample (127 mm × 13 mm × 13 mm) for 30 s and then removed. A series of five specimens of each composition were prepared. To calculate the burning rate, combustion time was recorded between the 25 mm and 100 mm marks. If the combustion front did not reach the 100 mm mark, the damaged length and elapsed time were recoded between the 25 mm mark and the point where the combustion front stopped. In case the burning rate did not exceed 40 mm/min over a 75 mm span for the specimen not thicker than 13 mm, or the combustion ceased before reaching the 100 mm mark, the material under investigation was rated as HB according to UL 94-HB classification.

## 3. Results and Discussion

The homogeneity of the nanoscale additive distribution within the volume of the foam is of significant importance, as it has a profound effect on the uniformity of the physical properties of the obtained composite PUF samples. To evaluate the homogeneity of TPP and HNT distribution in a polyurethane matrix, the elemental mappings of P, Si, and Al were employed. Figure 1A,B indicate that HNT and TPP are uniformly distributed within PUF/20HNT and PUF/20TPP, respectively. Figure 1C illustrates that the incorporation of the mixture of 20 wt.% HNT and 20 php TPP did not affect the distribution homogeneity of each additive. The same assumption can also be made for the mixture of NC and TPP.

The results of the investigation of the structure of the obtained foams are presented in Figure 2. According to the presented information, the incorporation of TPP results in a slight increase in PUF cell area, which is in good agreement with the measurements of the corresponding apparent densities. The addition of small amounts of the nanoscale additives (1 and 5 wt.%) did not result in a significant change to the foam density. Nevertheless, the cell area (i.e., cell volume) of the nanocomposites exhibited a slight decrease. The incorporation of 10–20 wt.% of nanoscale additives to the PUF led to an increase of foam apparent density and a decrease in the cell volume of the foam. It should be noted that the addition of a rather large amount of the nanoscale additive to the polyol led to an increase in its viscosity, which subsequently impeded the formation and growth of the gas bubbles during the foaming process. This finally resulted in a reduction in cell volume and an increase in foam density.

The polyurethane structures of the obtained reference PUF and modified foams were confirmed by FTIR spectroscopy (Figure 3). The bands in the 3000–2800 cm^−1^ region were attributed to C-H stretching, whereas the absorbance in the 1750–1660 cm^−1^ region was attributed to the stretching of carbonyl groups. The bands in the 1550–1500 cm^−1^ region originate from N-H in-plane bending vibrations, while the band at 1230 cm^−1^ originates from C-O stretching. The intense band at 1105 cm^−1^, as well as the group of bands in the 1000–850 cm^−1^ region, were attributed to the ester C-O-C symmetric stretching vibration [46,47,48,49].

The FTIR spectrum of TPP was represented by a few sharp bands in the range of 1700–800 cm^−1^. The bands at 1590 cm^−1^ and 1488 cm^−1^ were attributed to C=C stretching in the phenyl ring, while the band at 1292 cm^−1^ was attributed to P=O stretching. The bands in the 1260–1130 cm^−1^ and 900–660 cm^−1^ regions were assigned to C-H in-plane and out-of-plane oscillations, respectively, whereas the intense band at 952 cm^−1^ was due to P-O stretching [49]. The FTIR spectra of HNT and NC were found to be in good agreement with the literature data [42]: the characteristic bands of Si-O and Al-O vibrations and glucose ring were observed in the 1200–800 cm^−1^ region, respectively.

The FTIR spectra of the flame-retardant nanocomposite PUF presented the superposition of the bands listed above, indicating that there was no chemical interaction between TPP and the nanoscale additives and the polyurethane (and/or PUF starting components).

The incorporation of various compounds and additives may result in the undesired degradation of PUF’s mechanical characteristics. To verify the impact of TPP and natural nanoscale additives on PUF compressibility, corresponding investigations have been carried out (see Figure 4). As follows from the obtained data, the greater the quantity of TPP added, the greater the compressibility of the foam observed in comparison with the reference PUF. However, an increase in the amount of HNT or NC in the nanocomposite PUF leads to a subsequent decrease in the foam’s compressibility, which is in good agreement with the literature data [30,37,38,39,41]. Figure 4 indicates that NC introduction has a significantly greater influence on PUF compressibility than the introduction of the same amount of HNT. Summing up, one can conclude that the incorporation of the nanoscale additives in combination with TPP helps to counterbalance the negative impact on the foam’s mechanical characteristics of the latter.

The open-cell PUF has a low density and heat conductivity, which result in the rapid temperature rise of the foam’s surface when it is heated by a flame. Furthermore, the high oxygen permeability of open-cell foam makes oxidative reactions significant in flaming combustion [10]. In this regard, the investigation of the thermal stability of the reference PUF and the obtained nanocomposite foams in air was of great importance and interest.

Figure 5 presents the derivative thermogravimetry (dTG) curves of pristine foam, PUF modified with only TPP or nanoscale additive (HNT or NC), and nanocomposite PUF (PUF/TPP/HNT and PUF/TPP/NC). The shapes of the acquired dTG curves were found to be in good agreement with previous TGA observations of flexible PUF [10,47]. The decomposition of the reference foam involved a series of pyrolysis reactions (e.g., of polyurethane and regenerated polyol) and oxidative reactions (e.g., of polyurethane, regenerated polyol, and char) that occurred simultaneously over a similar temperature range and competed with each other. According to Figure 5, the weight of the reference PUF exhibited a pronounced decrease in a 200–400 °C temperature range. At 274 °C, pyrolysis of the foam took place, producing regenerated polyol and gaseous isocyanates. However, the polyol oxidation competed with this process and already reached its peak at 327 °C, releasing char, water, CO, CO_2_, CH_4_, and other substances. It was followed by polyol pyrolysis at 365 °C, giving even more char and gases. PUF decomposition was finished by char oxidation at 546 °C, resulting in final mass reduction via the emission of various gases.

The thermal decomposition of TPP occurs within a 200–350 °C range, which is consistent with the literature data [43] and, thus, overlaps with the PUF decomposition region. The modification of PUF by TPP did not change the starting temperature of its decomposition. However, the temperature range of the main mass decrease was significantly decreased. As expected, the larger quantity of TPP resulted in the most significant shrinking of this region. According to the literature review, such a pronounced mass decrease within a reduced temperature region indicates the accelerated formation of char, which acts as a protective layer against the further propagation of heat and flame [47]. It is also noteworthy that the peaks corresponding to the oxidation and pyrolysis of polyol overlapped to form a single broad peak. To interpretate this phenomenon, the hypothesis previously proposed was employed: the products of TPP thermal decomposition (which occurs in the same temperature region) catalyze foam and polyol oxidation, allowing it to happen at a lower temperature [10,46,48].

The introduction of HNT and NC jointly with TPP into the foam composition yielded different results, depending on the amounts of TPP and nanoscale additives. Thus, the small amounts of HNT and NC (1 wt.% and 5 wt.%) did not significantly influence the decomposition of the nanocomposite PUF. However, the addition of a greater quantity of additives (10 wt.% and 20 wt.%) resulted in a shrinking of the temperature region of the main mass drop or, in other words, led to a faster char formation in the 200–400 °C temperature range. This effect was particularly noticeable in the cases of the PUF/10TPP/HNT and PUF/10TPP/NC series, while in the cases of the PUF/20TPP/HNT and PUF/20TPP/NC series, the influence of the additive was prominent due to the rather high TPP content. Surprisingly, the samples with a high proportion of nanoscale additives and 10 php TPP demonstrated a faster rate of mass reduction (char formation) than samples with 20 php TPP.

The investigation of the thermal stability of the obtained nanocomposite PUF was followed by a flammability test. The UL 94 test, created in the Underwriters Laboratories, is the most common and widely used plastic flammability standard, determining whether the material is prone to either extinguishing or burning once it has been ignited. The combustion of the reference sample was completed within 47 ± 4 s (burning rate of 97 ± 7) mm/min) and was accompanied by extensive dripping. Consequently, the reference PUF can be considered as a highly flammable material.

The results of the flammability tests on the nanocomposite PUF are presented in Figure 6 and Table 2. The introduction of 10 and 20 php of TPP into the foam reduced its burning rate by 13% and 17%, respectively. However, when the error bars of the corresponding values are taken into account, the flammability levels of PUF/10TPP and PUF/20TPP are found to be very close to each other. Surprisingly, the modification of PUF with nanoscale additives (without TPP addition) resulted in a significant slowing of the burning rate of the obtained foams. The incorporation of 20 wt.% of HNT and NC resulted in a 41% and 48% reduction in the burning rate, respectively, in comparison to the reference PUF. The substantial enhancement in the fire resistance of PUF/20HNT and PUF/20NC can be related to the formation of a thicker char layer, which serves as a more effective heat shield, and to the endothermic breakdown of cellulose, which consumes the heat released during combustion for the thermal decomposition of a substance [2,9]). The thermal decomposition of cellulose occurs in the 200–400 °C region [42], which overlaps with the main mass reduction of polyurethane. It should be noted that a distinctive burnt sugar odor was observed during the combustion of the NC-containing foams, which supports our assumption about the cellulose antiflame activity resulting from its endothermic breakdown. Last but not least, unlike HNT, modification of PUF with both 10 and 20 wt.% of NC stopped the dripping of the melted polymer. However, the introduction of only nanoscale additives into the PUF did not allow the achievement of the HB classification according to the UL 94-HB standard, as the flame passed the 100 mm mark for all samples.

According to Figure 6 and Table 2, the introduction of even a small amount of the nanoscale additive (e.g., 1 and 5 wt.%) in combination with TPP allowed the substantial retardation of the burning rate of the obtained nanocomposite foams in comparison to the reference PUF, PUF/TPP, PUF/HNT, and PUF/NC. It should be underlined that the observed decreases in the burning rate for the PUF/TPP/HNT and PUF/TPP/NC series were not a result of the coexistence/addition of TPP and HNT/NC fire-retardant activities but of their interaction in a way that amplified their impact on the flammability of nanocomposite foam. A synergetic effect between various fire-retardant additives has been previously observed by many research groups [12,50,51,52,53,54,55,56,57,58].

However, it should be underlined that the shapes of the burning rate curves of the HNT- and NC-containing samples, shown in Figure 6, are different. Thus, one can note that the burning rates of PUF/10TPP/HNT drop as the HNT content increases from 0 to 5 wt.%, although the further addition of halloysite results in a gradual decline in foam nanocomposites flammability. Surprisingly, an increase of TPP content led to a less pronounced decrease in the burning rate for samples containing 5 wt.% and 10 wt.% of HNT. According to Figure 1, the distribution of TPP and HNT in the PUF/20TPP/5HNT and PUF/20TPP/10HNT samples was found to be homogeneous. Thus, one may suggest that the cell size and foam densities may be the cause of such unexpected burning rates. Indeed, an increase of PUF cell size leads to a greater degree of active gas convection and heat flow within the open-cell foam structure, which in turn supports its further combustion [9,59]. Figure 2 illustrates that the mean and median values of cell areas for PUF/20TPP/5HNT and PUF/20TPP/10HNT are greater than those observed for PUF/10TPP/5HNT and PUF/10TPP/10HNT. Additionally, the comparable flammability levels of the foams containing 15 and 20 wt.% of HNT in the PUF/10TPP/HNT and PUF/20TPP/HNT series can also be attributed to the similar values of the mean size of the cells in these samples. Unlike PUF modified with only nanoscale additives, no dripping of the melted polymer was observed in the case of the PUF/TPP/HNT samples containing 15 and 20 wt.% of halloysite. Finally, the PUF/10TPP/HNT samples containing 10 and 20 wt.% of halloysite and the PUF/20TPP/HNT samples with 5–20 wt.% halloysite content were rated as HB.

The incorporation of NC in combination with TPP into the PUF led to a rapid increase in the foam’s fire retardance. However, burning rates lower than 40 mm/min (allowing materials to be classified as HB) were observed only for the samples containing 10 wt.% of NC. For the interpretation of the obtained results, the cells’ area distribution was employed again. Indeed, the smallest cells of the PUF/10TPP/10NC and PUF/20TPP/10NC (see Figure 2) may play a key role in the foam’s combustion behavior. However, unlike PUF/TPP/HNT, variation of the TPP content in the PUF/TPP/NC samples did not significantly affect the flammability level of the nanocomposite foams. Moreover, Table 2 indicates that a minimum of 10 wt.% NC is required for PUF combustion without dripping.

Despite the comparatively low burning rates, no self-extinguishing of the nanocomposite PUF/TPP/NC samples was observed (with the exception of PUF/20TPP/10NC). To explore the difference between HNT and NC antiflame activity, the SEM and EDS investigations of the char residues of the PUF/20HNT, PUF/20NC, PUF/20TPP, PUF/20TPP/20HNT, and PUF/20TPP/20NC samples were performed (see Figure 7). Analysis of the SEM micrographs revealed that NC-containing samples formed rather porous char layers. Meanwhile, in the case of the samples with TPP and HNT, char layers were more uniform and denser. Such a difference between morphologies of the nanocomposite foam char layers could play a key role in flame propagation and extinguishing. Thus, for instance, the less dense and uniform char cannot completely exclude the diffusion of the flammable gases and heat within the undamaged foam volume. One more factor defining the extinguishing of the nanocomposite PUF samples could be the speed of char layer formation. According to our TGA observations (see Figure 5), the narrower region of the main mass drop of the PUF/TPP/HNT samples may indicate the faster formation of char than in case of the PUF/TPP/NC samples.

The investigations presented above demonstrate the existence of a synergetic effect between natural nanoscale additives (HNT or NC) and a commercially available fire retardant, TPP, allowing the significant enhancement of the fire resistance properties of the final nanocomposite PUF. The synthesis of the modified foams is simple and easily scalable. The incorporation of TPP in the mixture with natural nanoscale additives helps to overcome the negative effect of TPP on PUF’s mechanical properties. The employment of low-cost and environmentally friendly nanoscale additives paves the way for the development of a new generation of affordable fire-safe, porous materials, in compliance with the concept of sustainable development.

## 4. Conclusions

The high flammability of polyurethane foams still remains a significant concern. Thus, the development of new approaches to enhance their fire retardance in environmentally safe and cost-effective way is of the utmost importance. This study demonstrates the potential of the application of natural halloysite nanotubes (HNT) or nanocellulose (NC) in combination with a commercial, halogen-free fire retardant (TPP) in the synthesis of open-cell nanocomposite polyurethane foams.

The nanocomposite foams were obtained by dispersing a natural nanoscale additive and TPP prior to mixing with isocyanate. The ease of scalability and the possibility of producing foams of any shape and volume are the main advantages of the selected synthetic technique. The incorporation of TPP and additives slightly affects the microstructure and apparent density of nanocomposites compared to pristine foam. It was found that PUF’s compressibility increases after foam modification with TPP. Nevertheless, the addition of natural additives can overcompensate the negative effect of the latter.

The flammability tests revealed a notable synergetic effect between TPP and the natural additive (HNT or NC). Some of the obtained samples (e.g., PUF/TPP/20HNT and PUF/TPP/10NC) were found to comply with the UL 94-HB test and were thus rated as HB materials. It is important to note that the fire-retardant activities of the employed additives differ and depend on their nature. The high mechanical stability and low thermal conductivity of HNT facilitate the formation of a more reliable char layer, thereby preserving the undamaged foam from further heat and flame propagation. In contrast, NC provides its antiflame activity via endothermic breakdown. It has been demonstrated that the effect of microstructure on material fire retardance should not be underestimated, since the size of the foam cell volume affects heat flow and gas convection through the open-cell structure. Consequently, the content of additives and their impact on foam’s microstructure must be taken into account during the design process of fire-retardant nanocomposite foams.

## Figures and Tables

**Figure 1 polymers-16-01741-f001:**
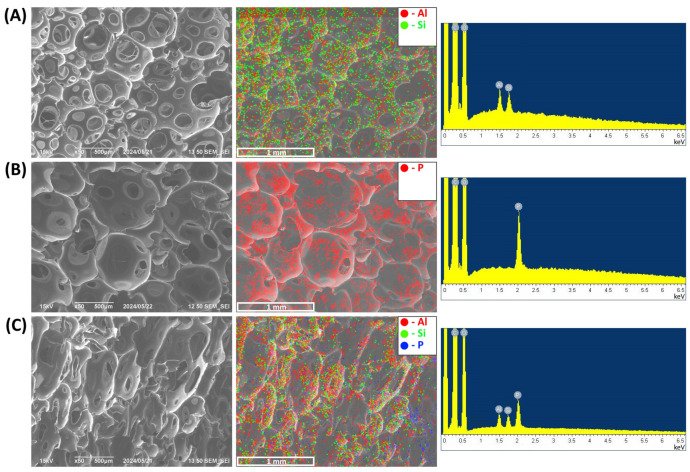
SEM micrographs, elemental mapping and the corresponding EDS spectrum of PUF/20HNT (**A**), PUF/20TPP (**B**), and PUF/20TPP/20HNT (**C**).

**Figure 2 polymers-16-01741-f002:**
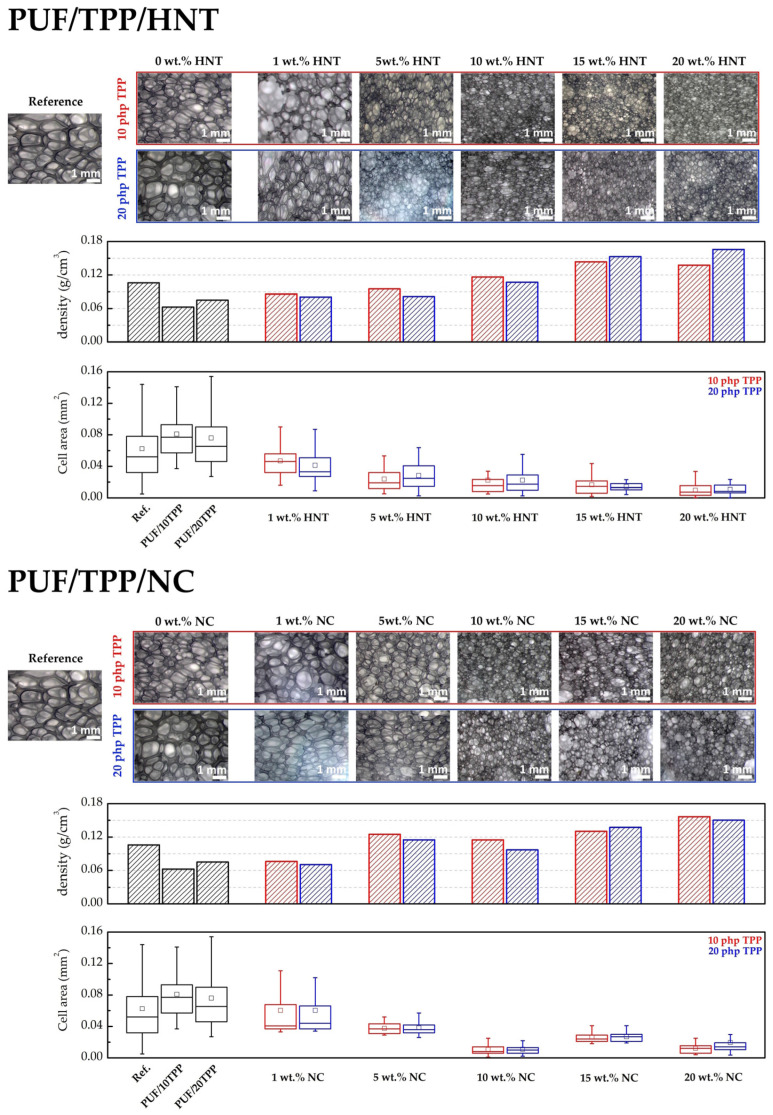
The photos of the cross sections, apparent densities, and box-plots of the cell area of PUF/TPP/HNT and PUF/TPP/NC.

**Figure 3 polymers-16-01741-f003:**
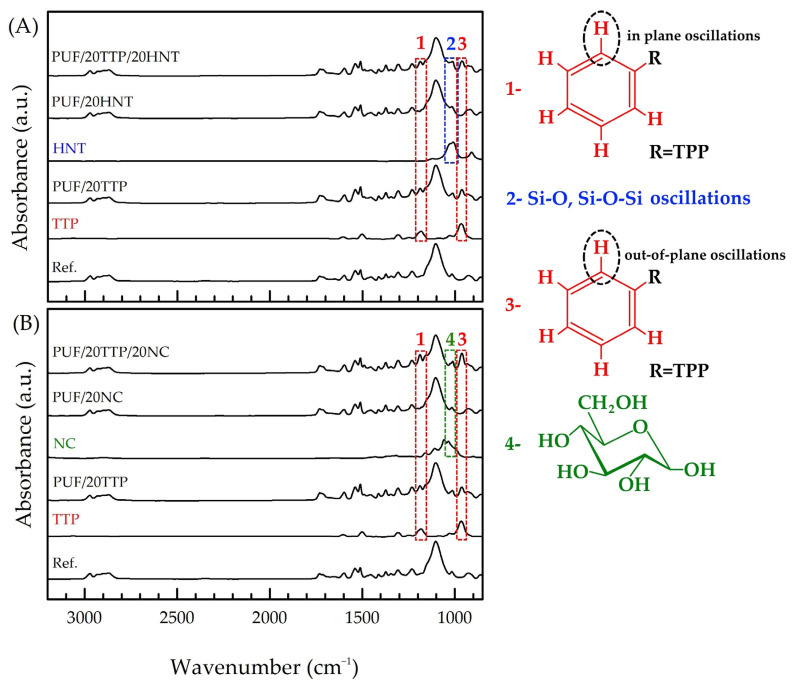
(**A**) FTIR spectra of individual HNT, TPP, reference PUF, PUF/20TPP, PUF/20HNT, and PUF/20TPP/20HNT; (**B**) FTIR spectra of individual NC, TPP, reference PUF, PUF/20TPP, PUF/20NC, and PUF/20TPP/20NC. The red, blue, and green dashed zones indicate the presence of TPP, HNT, and NC bands in the FTIR spectra of the final nanocomposite PUF samples, respectively.

**Figure 4 polymers-16-01741-f004:**
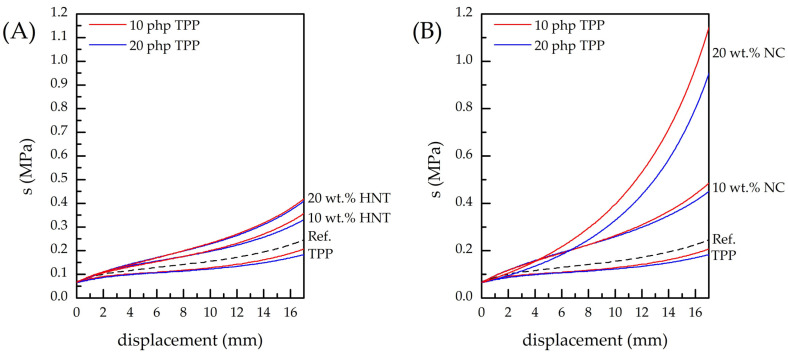
Results of compression force deflection measurements for the reference PUF, PUF modified with 10 php and 20 php of TPP and nanocomposite PUF: (**A**) PUF/10TPP/HNT and PUF/20TPP/HNT, (**B**) PUF/10TPP/NC and PUF/20TPP/NC.

**Figure 5 polymers-16-01741-f005:**
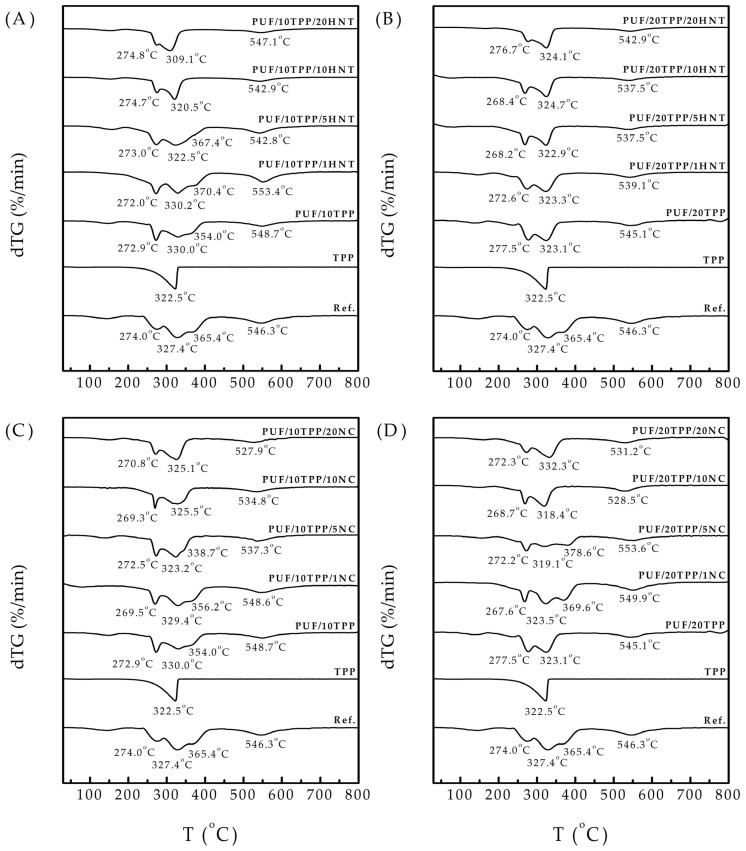
dTG curves of reference PUF, single TPP, PUF modified with 10 php and 20 php of TPP, and nanocomposite PUF: (**A**) PUF/TPP10/HNT, (**B**) PUF/TPP20/HNT, (**C**) PUF/TPP10/NC, and (**D**) PUF/TPP20/NC.

**Figure 6 polymers-16-01741-f006:**
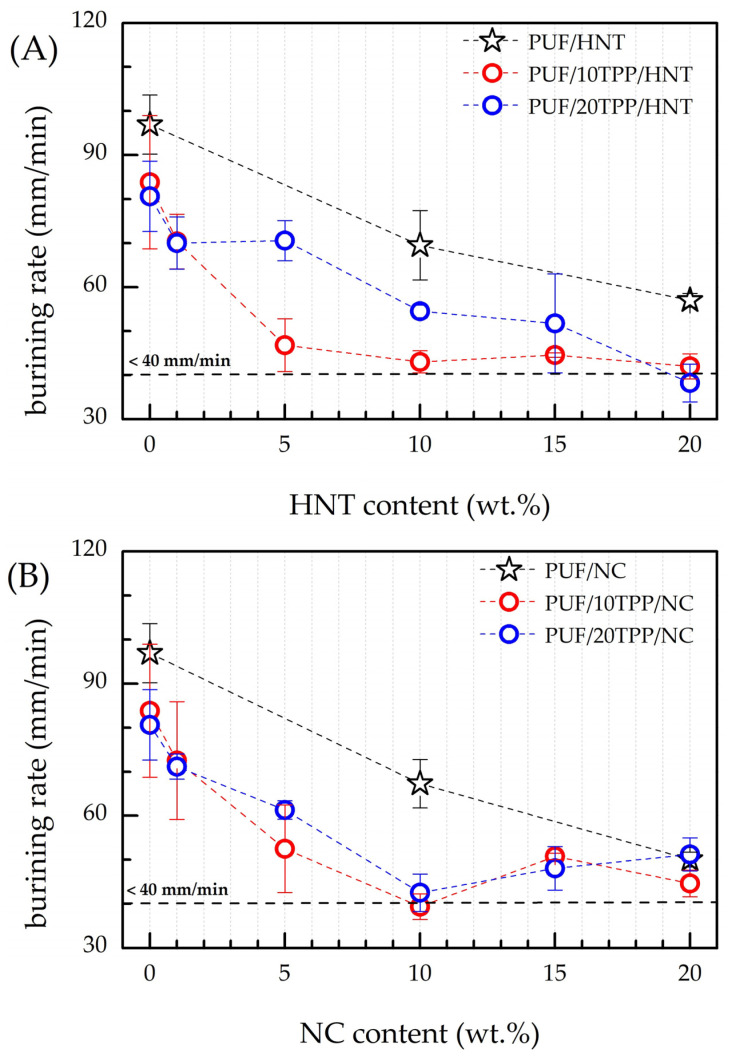
Burning rates of PUF/TPP/HNT (**A**) and PUF/TPP/NC (**B**) in comparison with reference PUF, foams modified with 10 and 20 php of TPP and 10 and 20 wt.% of HNT and NC. The horizontal dashed line represents burning rate of 40 mm/min, which corresponds to HB class.

**Figure 7 polymers-16-01741-f007:**
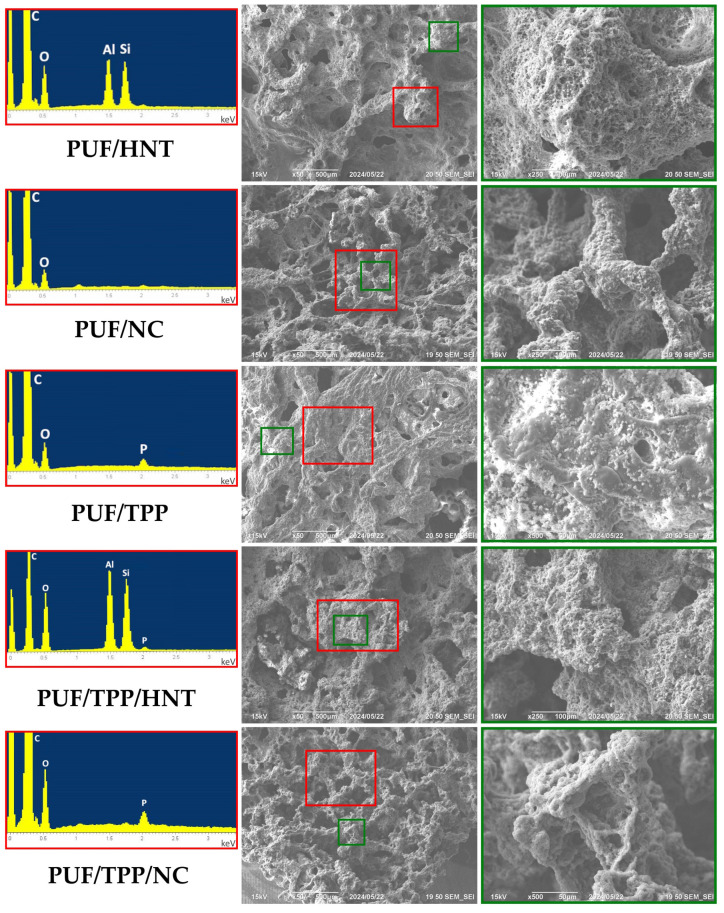
SEM micrographs of char layer and the corresponding EDS spectra of PUF/20HNT, PUF/20NC, PUF/20TPP, PUF/20TPP/20HNT, and PUF/20TPP/20NC. The zones of sample analyzed at higher magnification and zones of EDS analysis are marked by green and red, respectively.

**Table 1 polymers-16-01741-t001:** Formulations of synthesized nanocomposite PUF.

Specimen Name	TPP (php) *	HNT (wt.%)	NC (wt.%)
Ref.	0	0	0
PUF/10TPP	10	0	0
PUF/20TPP	20	0	0
PUF/10HNT	0	10	0
PUF/20HNT	0	20	0
PUF/10NC	0	0	10
PUF/20NC	0	0	20
PUF/10TPP/1HNT	10	1	0
PUF/10TPP/5HNT	10	5	0
PUF/10TPP/10HNT	10	10	0
PUF/10TPP/15HNT	10	15	0
PUF/10TPP/20HNT	10	20	0
PUF/10TPP/1NC	10	0	1
PUF/10TPP/5NC	10	0	5
PUF/10TPP/10NC	10	0	10
PUF/10TPP/15NC	10	0	15
PUF/10TPP/20NC	10	0	20
PUF/20TPP/1HNT	20	1	0
PUF/20TPP/5HNT	20	5	0
PUF/20TPP/10HNT	20	10	0
PUF/20TPP/15HNT	20	15	0
PUF/20TPP/20HNT	20	20	0
PUF/20TPP/1NC	20	0	1
PUF/20TPP/5NC	20	0	5
PUF/20TPP/10NC	20	0	10
PUF/20TPP/15NC	20	0	15
PUF/20TPP/20NC	20	0	20

* php—parts per hundred polyol (component B) by weight.

**Table 2 polymers-16-01741-t002:** The results of UL 94-HB test.

Specimen	V (mm/min) Mean ± Std	Dripping	Flame Passed 25 mm Mark	Flame Passed 100 mm Mark	Rating
Ref.	97 ± 7	Yes	Yes	Yes	—
PUF/10TPP	84 ± 15	Yes	Yes	Yes	—
PUF/20TPP	81 ± 8	Yes	Yes	Yes	—
PUF/10HNT	70 ± 8	Yes	Yes	Yes	—
PUF/20HNT	57 ± 2	Yes	Yes	Yes	—
PUF/10NC	67 ± 5	No	Yes	Yes	—
PUF/20NC	50 ± 2	No	Yes	Yes	—
PUF/10TPP/1HNT	70 ± 6	Yes	Yes	Yes	—
PUF/10TPP/5HNT	47 ± 6	Yes	Yes	Yes	—
PUF/10TPP/10HNT	**43** ± **3**	Yes	Yes	Yes	**HB**
PUF/10TPP/15HNT	45 ± 1	No	Yes	Yes	—
PUF/10TPP/20HNT	**42** ± **3**	No	Yes	**No**	**HB**
PUF/20TPP/1HNT	70 ± 6	Yes	Yes	Yes	—
PUF/20TPP/5HNT	71 ± 5	Yes	Yes	**No**	**HB**
PUF/20TPP/10HNT	55 ± 1	Yes	Yes	**No**	**HB**
PUF/20TPP/15HNT	52 ± 11	**No**	Yes	**No**	**HB**
PUF/20TPP/20HNT	**38** ± **4**	**No**	Yes	Yes	**HB**
PUF/10TPP/1NC	73 ± 13	Yes	Yes	Yes	—
PUF/10TPP/5NC	53 ± 10	Yes	Yes	Yes	—
PUF/10TPP/10NC	**39** ± **3**	**No**	Yes	Yes	**HB**
PUF/10TPP/15NC	51 ± 1	No	Yes	Yes	—
PUF/10TPP/20NC	45 ± 3	No	Yes	Yes	—
PUF/20TPP/1NC	71 ± 3	Yes	Yes	Yes	—
PUF/20TPP/5NC	61 ± 2	Yes	Yes	Yes	—
PUF/20TPP/10NC	**43** ± **4**	**No**	Yes	**No**	**HB**
PUF/20TPP/15NC	48 ± 5	No	Yes	Yes	—
PUF/20TPP/20NC	51 ± 4	No	Yes	Yes	—

## Data Availability

The original contributions presented in the study are included in the article, further inquiries can be directed to the corresponding author.

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
