# Peer review of "New Fire-Retardant Open-Cell Composite Polyurethane Foams Based on Triphenyl Phosphate and Natural Nanoscale Additives"

_polymers, 2024, doi:10.3390/polym16121741_

Round 1

Reviewer 1 Report

Comments and Suggestions for Authors

1. Some parts of the manuscript are not adequately presented, so the authors must check the whole text to optimize the language presentation. For example, “Among a great variety of researches performed over the last decades to enhance PUF fire resistance [9,12–16], three main approaches can be distinguished: (1) chemical modification of polyurethane chain (e.g., introduction of fire resistant fragments/functional 36 groups); (2) fire resistant coating of the PUF surface (e.g., via LbL technique or foam soaking); (3) introduction of various flame retardants and other materials (including nano- materials) during PUF synthesis. The latter approach implies addition of the additive in one of the PUF components (usually polyol) before mixing with the second component and further foam formation.”

2. In Figure 1, the authors note the apparent density and box plots of foam cell areas. Could the authors have added scanning electron microscopy images to more visually corroborate the apparent map of foam cells in the study?

3. The manuscript contains fewer tests and data on the flame-retardant properties of the composites, and the authors should employ more extensive testing to measure their flame-retardant properties.

4. Compared to previous related works, whether the TPP/(HNT and NC) have significant advantages in flame?

5.  Figure 2 shows Fourier Transform Infrared Spectroscopy (FTIR), which is not consistent with this statement “As it follows from see Figure 2 the nanocomposite foams with 10 wt.% of NC possess the smallest cells, which may play a key role in their combustion behavior.”. The authors are requested to make appropriate corrections and pay attention to the details of the manuscript presentation.

Comments on the Quality of English Language

The English of total manuscript should be checked and improved.

Reviewer 2 Report

Comments and Suggestions for Authors

The manuscript by Anton Semenov and coworkers demonstrates the potential of application of natural nanomaterials (HNT and NC) in combination with com- mercial halogen-free fire retardant (TPP) in synthesis of open-cell nanocomposite polyurethane foams. Despite the exciting results of the study, the authors did not give enough emphasis on the advantages of using the domain-limited orientation approach. In addition, some points should be clarified and added.

1.      How HNT and TPP are dispersed in the substrate? Could the author give some direct evidence?

2.      10 wt.% of NC/20 wt.% of HNT and TPP are benefit for the flame-retardancy properties by the burning testing. Could you provide some more quantitative data, like cone, UL-94 etc?

3.      The author should modify the grammar and expression very carefully.

Comments on the Quality of English Language

The author should modify the grammar and expression very carefully.

Reviewer 3 Report

Comments and Suggestions for Authors

In this work, a flame-retardant PU foam with halloysite and nanocellulose was studied. The current manuscript needs major revisions before acceptance and some recommendations are listed.

1.      What is the novelty of your work? Why do you want to use Halloysite and NC biopolymers? These two are common flame-retardant additives.

2.      The Specimen Names are unreadable in Table 1, please modify it.

3.      For Figure 1, it is better to add titles to both figures, it is hard to read.

4.  In Figure 2, it is better to provide the group information near the dashed box you marked.

5.      In Figure 4, what is the burning rate? It is better to provide the calculation information.

6.      In the conclusion part, “This study demonstrates the potential of application of natural nanomaterials (HNT and NC) in combination with commercial halogen-free fire retardant (TPP) in the synthesis of open-cell nanocomposite polyurethane foams.”

Why did you mix TPP with HNT and NC respectively, not mix all of them, or mix HNT and NC? From this sentence, it would like you to use HNT and NC together as flame retardant additives. 

Comments on the Quality of English Language

 Minor editing of the English language required

Reviewer 4 Report

Comments and Suggestions for Authors

This work aims to show the feasibility of successful application of such natural nanomaterials as halloysite nanotubes and nanocellulose as promising additives to commercial halogen-free fire retardant, triphenyl phosphate, in order to enhance the flame retardance of the open cell polyurethane foams. The article can be accepted after undergoing major revisions.

1: This work lacks research and discussion on the flame retardant mechanism. A systematic analysis should be conducted on the condensed and gaseous products of combustion to analyze the flame retardant mechanism of halogenated minerals, triphenyl phase, and nanocellulose.

2. What is the compatibility between halloysite, triphenyl phase, and nanocellulose and the matrix? Will halloysite undergo migration and precipitation, and should surface modification be carried out on halloysite?

3. In the abstract, the author mentioned "Describe mechanical and physical properties of foam their application is still hidden by high inflammability", but in this study, the author did not study the mechanical properties of foam? How does the use of halloysite, triphenyl phase, and nanocellulose affect mechanical properties?

Comments on the Quality of English Language

 Moderate editing of English language required

Round 2

Reviewer 1 Report

Comments and Suggestions for Authors

1. Figure 2 title labeling is not clear, it is recommended that the title labeling be revised.

2. There is no scale labeling on the SEM images in Figure 2, which creates visual difficulties for reviewers and readers.

Reviewer 2 Report

Comments and Suggestions for Authors

The authors have revised the MS properly which can be accepted.

Reviewer 3 Report

Comments and Suggestions for Authors

Accept in this version

Reviewer 4 Report

Comments and Suggestions for Authors

Minor editing of English language required

Comments on the Quality of English Language

Minor editing of English language required